# Highly Branched Polymers Based on Poly(amino acid)s for Biomedical Application

**DOI:** 10.3390/nano11051119

**Published:** 2021-04-26

**Authors:** Marisa Thompson, Carmen Scholz

**Affiliations:** Department of Chemistry, University of Alabama in Huntsville, 301 Sparkman Dr., Huntsville, AL 35899, USA; met0021@uah.edu

**Keywords:** poly(amino acid), dendrimer, dendrigraft, hyperbranched polymer, gene delivery, drug delivery, antiviral

## Abstract

Polymers consisting of amino acid building blocks continue to receive consideration for biomedical applications. Since poly(amino acid)s are built from natural amino acids, the same building blocks proteins are made of, they are biocompatible, biodegradable and their degradation products are metabolizable. Some amino acids display a unique asymmetrical AB2 structure, which facilitates their ability to form branched structures. This review compares the three forms of highly branched polymeric structures: structurally highly organized dendrimers, dendrigrafts and the less organized, but readily synthesizable hyperbranched polymers. Their syntheses are reviewed and compared, methods of synthesis modulations are considered and variations on their traditional syntheses are shown. The potential use of highly branched polymers in the realm of biomedical applications is discussed, specifically their applications as delivery vehicles for genes and drugs and their use as antiviral compounds. Of the twenty essential amino acids, L-lysine, L-glutamic acid, and L-aspartic acid are asymmetrical AB2 molecules, but the bulk of the research into highly branched poly(amino acid)s has focused on the polycationic poly(L-lysine) with a lesser extent on poly(L-glutamic acid). Hence, the majority of potential applications lies in delivery systems for nucleic acids and this review examines and compares how these three types of highly branched polymers function as non-viral gene delivery vectors. When considering drug delivery systems, the small size of these highly branched polymers is advantageous for the delivery of inhalable drug. Even though highly branched polymers, in particular dendrimers, have been studied for more than 40 years for the delivery of genes and drugs, they have not translated in large scale into the clinic except for promising antiviral applications that have been commercialized.

## 1. Introduction 

The synthesis of highly branched poly(amino acid)s, PAAs, dates back to the late 1950s, when it was observed that amino acids could be thermally polymerized [1,2,3,4]. Amino acids with functional side groups proved to be more readily polymerizable and early on Lysine was one of the focal points. However, structural intricacies of the resulting condensation products were not known at the time, and the products were described as “gel-like” or “sponge-like” [1]. Rohlfing even went as far as connecting the thermal formation of poly(amino acid)s to prebiotic proteins on a prebiotic Earth [5].

More recently, highly branched PAAs gained attention for their ability to (i) mimic globular proteins, so called ‘proteinoids’, which can be studied as mock proteins or peptides for their interaction with various environments [6,7,8,9] and were mentioned as early as 1958 in this capacity by Fox and Harada [10], and (ii) interact with biologically relevant molecules. The latter one has become the more intensely researched area as PAAs lend themselves to a multitude of biomedical applications, including the delivery of drugs [11,12,13] and nucleotides, for a review see Kataoka [14], and as imaging reagents, [15,16] and tissue adhesives [17,18] and most recently in the development of theranostics [19,20]. Applications in the aforementioned fields are based on the highly branched PAAs’ ability to self-assembling into nanosized structures, thereby matching a size-range that is demanded for applications in biomedicine. Three types of highly branched PAAs can be distinguished: (i) dendrimers, (ii) dendrigrafts and (iii) hyperbranched polymers, Figure 1. These structures are built from asymmetrical AB_2_ monomers that provide the bi-functionality necessary for the successive building of generations and formation of branched structures. 

The synthesis of dendrimers, sometimes referred to as arborescent polymers [21,22] is the most challenging as only one monomer is added per branching point and step, and deprotection and purification steps are necessary before the next generation can be added. Compared to dendrigrafts and hyperbranched polymers, dendrimers are the most defined structures, and the number of terminal functional groups, or charge density per polymer is pre-determined. Dendrigrafts are also built-in generations but in a less controlled manner. Starting from a linear polymer, the terminal functional side groups act as initiators for the formation of the second generation, which in itself is another short linear polymer. While the second generation of a dendrigraft could also be referred to as a comb-type polymer, the third generation is then built from the terminal functional side groups of the second and, potentially, also first generation. All polymeric generations are linear polymers that can vary in lengths. Deprotection and purification steps are necessary after the formation of each generation. The synthesis of hyperbranched polymers is typically performed in a one-step thermal polymerization, which proceeds with no control over the resulting polymer structure. The synthesis of hyperbranched polymers is comparatively simple, and large quantities can be prepared rather fast, as iterative purification steps that are necessary for generation-based polymers are omitted. Unlike dendrimers, dendrigrafts and hyperbranched polymers are not monodisperse and display structural defects. 

## 2. Poly(amino acid)-Based Dendrimers

Dendrimers are unique scaffolds in that their size is adjustable based on the generation number and that a pre-determined number of functional groups is available for the attachment of molecules that determine the ultimate purpose of the construct. Pendant molecules attached to dendrimer scaffolds are in close proximity to one another, basically forming the outermost layer of a spherically structure, provided complete or nearly complete coupling can be achieved. This is in distinct contrast to side-chain modified linear molecules. PAA-based dendrimers have an inherent advantage, over other, e.g. poly(amidoamine) and poly(propylene imine) [23] dendrimers; they are susceptible to protease degradation and the degradation products are amino acids that, once they are released from the supermolecular dendritic parent structure, can enter into biochemical pathways. 

The controlled synthesis of PAA dendrimers is built upon the pioneering work of Bayer and Mutter who advanced Merrifield’s solid phase peptide synthesis method and developed a liquid-phase peptide synthesis, which overcame the problem of incomplete coupling [24,25]. In their ingenious work they suggested poly(ethylene glycol), PEG, as useful solubilizing moiety in addition to poly(vinyl alcohol) and poly((vinylamine)-co-(vinylpyrrolidone)). Bayer and Mutter’s original concept, which targeted the formation of linear peptides with a controlled sequence that are easy to isolate at each step, was later adopted to the liquid-phase synthesis of PAA dendrimers.

Even though dendrimers are polymeric structures, their syntheses resemble more those of small molecules as only one monomer is added to an initiating functional group per reaction step and no chain growth occurs [26]. In the case of poly(*L*-lysine), p(*L*-Lys), dendrimers, *N*-Boc- or *N*-Fmoc-protected *L*-Lysine is coupled via an amide bond to an initiator core carrying a certain number of amino groups. After deprotection, the first generation of the dendrimer is completed and now displays two amino-groups for each one present in the initiator. These amino groups are now available for the coupling of two more protected *L*-lysine molecules per initiating amino function. Thus, the number of reactive sites doubles in each generation. These coupling reactions are typically base-catalyzed and often use activating reagents. The draw-back of this method is that the reaction has to be stopped after the addition of one molecule per reactive site, i.e., after the formation of each generation and a laborious deprotection and purification procedure ensues. Figure 2 shows the formation of a 3rd generation dendrimer from an initiator with a single amino group; 16 reactive sites are available at this 3rd generation dendrimer. At higher generations significant crowding occurs at the reactive sites, which can lead to incomplete conversions at higher generations.

### 2.1. Poly(L-Lysine)-Based Dendrimers

Lysine plays a key role as building block for dendritic PAAs; it is an asymmetrical AB_2_ monomer that is uniquely suited for the formation of p(*L*-Lys) dendrimers. Amino-terminated p(*L*-Lys) dendrimers are of interest as their corona is positively charged at physiological pH. Catalyzed by the Human Genome Project nucleic acids, specifically plasmid DNA and RNA, are now considered a class of therapeutics, for which delivery vectors are needed. Hence, functions involving the delivery of genes that stimulate the synthesis of a therapeutic protein in the patient’s cells or the silencing of genes that overproduce proteins or miscoded proteins present themselves as prominent application of p(*L*-Lys) dendrimers. Dendritic structures with cationic surfaces interact, bend and destabilize anionic cell membranes [27,28] thus, ingress into the cell is achieved, known as extravasation. Similar mechanisms play a role when entering the nucleus. 

A structural variation to the dendrimer concept described above was introduced by Heise and Cryan [29] who controlled the number of dendritic arms by using 2nd to 5th generation poly(propylene imine) dendrimers as macroinitiators. Here, a 2nd generation poly(propylene imine) carries eight amino groups, a 5th generation carries 64 amino groups. Each of these amino groups initiates the ring-opening polymerization, ROP, of ε-carbobenzyloxy-*L*-Lysine N-carboxyanhydride. As these ROPs typically follow a living or pseudo-living mechanism the number of repeat units in the resulting arms can be expected to be close to the target chain length as pre-determined by the monomer-to-initiator ratio. Hence, in the resulting star-shaped polymeric structures the number of p(*L*-Lys) arms and their chain length is pre-determined, Figure 3. These linear arms are synthesized in a single step; dendritic poly(propylene imine) macroinitiator are commercially available. 

### 2.2. Delivery of Nucleic Acids Using p(L-Lys) Dendrimers

Research into dendritic structures used in gene delivery has been summarized before [30,31,32]. Here, selected examples will be discussed that highlight the impact of the structure-property relationship on the use and effectiveness of p(*L*-Lys)-based dendrimers. 

Park et al. applied Bayer and Mutter’s classic approach for peptide syntheses in solution by keeping the growing peptide solubilized via a polymeric anchor [33], Figure 2. The authors showed that the 4th generation dendrimer is capable of completely condensing pDNA at an N:P ratio of 2:1 by forming nanoparticles of about 100 nm, which were stable against DNase. The authors expanded on this concept by generating dumbbell shaped constructs using a homo-bifunctional PEG to initiate the liquid phase peptide synthesis [34], Figure 2. These early studies focused on dendrimer syntheses and the condensation of nucleic acids and did not test the transfection efficiency. 

In the course of studying different molecular architectures, it was discovered that DNA condensation efficiency depends on the dendrimer generation, that is, charge density, and to a lesser extent on the N:P ratio. A 3rd generation copolymer was not able to completely complex DNA, even at high (20:1) N:P ratios, however, the increase in cationic charge density in the 4th generation yielded complete complexation and protection against DNase activity at an N:P ratio as low as 1:1 [34]. Aoyagi et al. using hexamethylene as initiator confirmed these observations and the importance of charge density by showing that low generation p(*L*-Lys) dendrimers do not condense DNA; at least a 4th generation dendrimer is necessary for acceptable complexation, and 5th and 6th generation dendrimers showed even better complexation and higher transfection abilities [26,35]. The dendrimer core seemed to have less of an impact on transfection efficiency, but as shown later can have an impact on the overall toxicity of the construct. 

As expected, the star-shaped polymeric constructs introduced by Heise and Cryan readily formed polyplexes with pDNA and siRNA [29]. Compared to linear p(*L*-lys) the star shaped polymers performed superiorly, producing tightly packed polyplexes at significantly lower N:P ratios. Yet, a distinct impact of the star-polymer architecture on the nature of the polyplexes was observed. Star-shaped polymers with fewer arms, i.e., formed from a lower generation of macroinitiators, yielded structures where the p(*L*-Lys) branches are less densely packed. Those were the more promising pDNA and siRNA delivery vectors. It was found that the number of cationic functional groups is not solely responsible for the complexation of pDNA and siRNA, but how these cationic charges are presented. A large dendrimer core that produced a large number of tightly packed p(*L*-Lys) arms led to a limited interaction with the nucleotides, probably restricted to the periphery of the spherical construct. On the other hand, structures with fewer arms, which have an overall lower charge density, but more flexibility, seemed to allow for increased interaction between the polymeric star-shaped vector and the nucleic acid cargo. It can be assumed that nucleic acids, especially the smaller siRNAs can “crawl” better into the dendrimer when there is more flexibility in the p(*L*-Lys) arms. Successful transfection of epithelial cells was shown for these systems.

The group of Klajnert-Maculewicz [36,37] expanded on the concept of p(*L*-Lys) dendrimers for the delivery of siRNA by incorporating arginine and histidine residues into the dendritic structure [38], Figure 4. Such a substitution will impact the flexibility of the dendrimer, the charge density and the eventual endosomal escape and may alter the way the delivery system interacts with nucleic acids. No significant difference in cellular uptake of dendrimer-siRNA complexes was observed for the three types of dendrimers: *L*-histidine- and *L*-arginine-augmented and exclusively *L*-lysine based dendrimers. However, differences in the toxicity towards myeloid cell lines were observed that followed the differences in the zeta potentials of the structures. The pure p(*L*-Lys) structure with the highest positive surface charge showed the highest toxicity. The *L*-histidine-augmented structure on the other hand had the lowest charge density and also displayed the lowest toxicity. This finding confirms the well-established mechanism of cell death caused by dendrimers with a large positive charge density. 

### 2.3. Delivery of Therapeutics Using p(L-Lys) Dendrimers

Considering the literature of the last five years, reports on the use of p(*L*-Lys) dendrimers in gene delivery applications have become sparse. This can be attributed to (i) the laborious synthesis and (ii) transfection results achieved by dendritic structures do not justify the time and effort necessary to prepare these delivery vehicles. More recent research has shifted to drug delivery systems, first and foremost the delivery of anticancer drugs [39,40,41,42,43] and contrast and imaging agents [44,45]. 

In order to design and synthesize efficient drug delivery systems, it is necessary to understand the uptake, internalization and interaction of the dendritic structure with cellular components. Avaritt and Swaan [46] showed that there is a multitude of pathways for the uptake of dendrimers. Using a human colorectal adenocarcinoma cell line, they showed that p(*L*-Lys) dendrimers utilize a variety of uptake mechanisms, including caveolin-, cholesterol-, and dynamin-mediated endocytosis, but the cholesterol-mediated uptake and micropinocytosis are the preferred mechanisms. 

An important finding with respect to drug delivery was that internalization and trafficking of the dendrimers within cells was significantly impacted by the modes of conjugation of a drug or drug surrogate onto the dendrimer. Avaritt and Swaan [46] attached a hydrophobic drug mimic, specifically a fluorophore, either to the dendrimer corona or the dendrimer core, and showed that the conjugation site determined the dendrimer toxicity. The dendrimers showed lower levels of cell toxicity when the fluorophore was attached to the dendrimer corona thereby partially blocking the surface amino groups and reducing the overall toxicity. When the fluorophore was attached to the core it was assumed that an amphiphilic dendrimer formed that was able to undergo dimerization, and shelter the fluorophore in a hydrophobic pocket. Formation of such pockets had been shown before by Niidome et al. [47]. This dendritic dimer will present a high concentration of amino groups on its surface, thus causing increased toxicity. Thus, the way drugs are conjugated to a (p(*L*-Lys) dendrimer plays a significant role in the behavior and efficacy of the drug delivery system, showing that such systems need to be judiciously designed to achieve optimum results. 

Incorporated fluorophores can provide several advantages to dendritic systems used in cancer treatments. Starting from a perylene diimide fluorophore, 8th generation macroinitiator p(*L*-Lys) dendrimers were synthesized by Zhou et al. [45] with the fluourophore forming the dendrimer core similar to the work by Avaritt and Swaan [46] discussed above. The fluorophore brightness was tunable, and it intensified with dendrimer size. Because the fluorophore was sheltered inside a dendritic pocket it had better fluorescence quantum yields, longer fluorescence lifetimes, and an improved in vitro and in vivo photostability compared to other molecular fluorescent probes without a polymeric shell. Serving emerging theranostics concepts, these fluorophore containing dendrimers will not only be useful as drug delivery systems but also show considerable promise for cancer imaging and image-guided surgery. Using such constructs will (i) aid in diagnosing tumors, provided they are equipped with the appropriate homing device on the dendrimer surface that guides the dendritic structure to a tumor site, (ii) guide the complete surgical removal of a tumor, as the fluorescence is readily observable and (iii) since there will be residual dendrimers left in the tumor vicinity after surgery, chemotherapeutics can be delivered to the site to guarantee the destruction of lingering cancer cells. 

The group of Kaminskas has contributed significant insight into the use of p(*L*-Lys)-based dendrimers as drug delivery systems in general [48,49,50,51,52] and inhalable systems in particular [53,54,55,56,57]. The inhalability of dendrimer-based delivery systems is of particular interest to the treatment of lung diseases. Inhalable dendritic p(*L*-Lys) drug delivery systems have shown superior activity as compared to administering the same drug intravenously. Lung tissue is very sensitive to particulates, including those left behind from drug delivery systems. Typical liposome or nanoparticulate delivery systems with diameters between 50–600 nm are too big for inhalable applications as these particulates pose significant toxicity risks because they will infiltrate alveolar macrophages and cause inflammation. Hence, delivery systems were sought that are much smaller and would not pose a toxicity risk. Dendritic structures, which can be as small as 20 nm in diameter, match these requirements and are advantageous with respect to interstitial diffusion, penetration into a tumor and the extent of absorption achieved from these delivery systems.

Moreover, synthesizing the dendrimer from an amino-terminated PEG macroinitiator results in dendritic structures with the added advantage of being readily absorbed into the blood stream. Thereby the build-up of particulate carriers in the lung tissue is kept low. As the carrier, potentially still being loaded with a residual amount of drug, diffuses into the blood stream treatment of the tumor from the ‘air-side’ as well as ‘blood-side’ becomes possible [55]. Aside from their value in the special case of inhalable drugs, the use of PEGylated p(*L*-Lys) dendrimers showed all the same advantages that are known for other PEGylated delivery systems, such as being long-circulating vectors [49], having a lower systemic toxicity [50] and being useful for the chemotherapeutic treatment of the lymphatic system [51]. 

### 2.4. Delivery of Antiviral Compounds Using p(L-Lys) Dendrimers

Dendrimers have been used as structural cores for the formation of antiviral compounds. Antiviral activity has been found in dendritic PAAs where the corona was derivatized with saccharide moieties, i.e., glycodendrimers. Typically, p(*L*-Lys) dendrimers were used, Figure 5. Uryu’s group tested a variety of saccharides and oligosaccharides, including lactose, maltose, maltotriose, glucose and acetyl cellbiose. These oligosaccharides were connected to a p(*L*-Lys) dendrimer by reductive amination between the sugar reducing end and the terminal amino group of the dendrimer, often via an amino acid linker such as alanine or leucine. Higher generation p(*L*-Lys) dendrimers, i.e., dendrimers with a large number of amino groups available for functionalization, showed higher antiviral activity when their surfaces were decorated with any of those sugar moieties [58,59,60,61]. Sufficiently high anti-HIV activities were found at low dendrimer concentrations, and the constructs showed comparatively low cytotoxocity. This research was taken to another level when an HIV-specific peptide sequence, the V3 loop (third variable region) of the envelope gp120 glycoprotein, was attached to the saccharide corona of the dendrimer. This peptide sequence is known to induce antibody formation, which then neutralizes viruses that express the same or a similar peptide sequence in their envelope structure [62]. For further reading on the use of dendrimers in glycobiology the reader is referred to a review by Roy and Touaibia [63]. 

Starpharma, an Australian biopharmaceutical company has taken dendrimer-based antivirals to market, focusing on HIV-preventive products. Given the 2020 coronavirus pandemic, researchers have looked to Starpharma’s dendrimer platform SPL7013, Figure 6. SPL7013 is a 4th generation p(*L*-Lys) dendrimer with broad spectrum antimicrobial activity, against several enveloped and non-enveloped viruses [64,65,66], and is marketed for antiviral and antibacterial applications. SPL7013 was synthesized from an initiator that consists of the benzhydrylamine amide of *L*-Lysine. The 32 amino groups present at the surface of this dendrimer were derivatized with sodium 1-(carboxymethoxy) naphthalene-3,6-disulfonate, thus generating a p(*L*-Lys) dendrimer with an anionic surface charge, with the commercial name astodrimer sodium. Paull et al. evaluated the antiviral activity of SPL7013 and found that it inhibits replication of SARS-CoV-2 in Vero E6 cells when added to cells one hour prior to, or one hour post infection [67], in a way similar as observed previously for the inhibition of HIV [68]. Since it is a dendrimer-based antiviral compound, inhalability is possible, and the authors conclude that SPL7013 warrants investigation for nasal administration to aid in preventing viral transmission and that it could be useful in the treatment of SARS-CoV-2. As in other applications of SPL7013, the compound proved to be a potent inhibitor of early events in the virus lifecycle. It is suggested that the antiviral activity is based upon strong electrostatic binding to positively charged amino acid clusters on the virus thereby inactivating it and blocking infection. 

## 3. Poly(*L*-Glutamate), p(*L*-Glu)-Based Dendrimers

The body of work on p(*L*-Glu)-based dendrimers is less extensive than that for p(*L*-Lys)-based dendrimers. Glutamate is also an AB_2_ asymmetrical monomer, here the carboxyl groups form the branching points, and dendrimers are formed by the successive formation of amide linkages between the two peripheral carboxyl groups in the dendrimer structure and the amino group of the incoming glutamic acid monomer. Rather than building dendrimers from *L*-Glu, this amino acid has been used to amend dendrimers, mostly based on poly(amidoamine)s [69], that were considered for drug or gene delivery [70]. 

Vinogradov et al. exploited p(*L*-Glu) dendrimers to control the local environment of catalytic structures, specifically porphyrin. The p(*L*-Glu) dendrimer generated a microenvironment for porphyrin cores that mimicked heme-containing proteins. The construct can be fine-tuned and was used to regulate and study the access of small molecules to the catalytic core. Starting from a Pd porphyrin core with terminal carboxyl groups a p(*L*-Glu) dendrimer was built to several generations. In an effort to alter the chemical environment, a hydrophobic shell was introduced by reacting the dendrimer with aminoundecanoic acid. The terminal carboxyl group of the fatty acid was then used to add an additional glutamate layer. The microenvironment of the dendritic structure can be influenced by the chemical nature of the intermittent hydrophobic layer. Solvent effects that resulted from the interaction of the intermittent layer showed shrinking of the construct when in contact with water and swelling in the presence of DMF. More densely packed structures restricted the diffusion of oxygen, which was monitored via phosphorescence quenching constants [71,72].

Another approach to the incorporation of a porphyrin into p(*L*-Glu) dendrimer was provided by Hartwig et al. [73]. The p(*L*-Glu) dendrimer was built by a less-common accelerated synthesis, that is, iterative divergent/convergent exponential growth [74]. Starting from a dipeptide, generation 1—here diglutamate—employing a specific protection/deprotection chemistry, generation 2 was prepared by the attachment of two selectively deprotected glu-dipeptides fragments to the generation 1 core. Generation 4 is then prepared by the attachment of two selectively deprotected generation 2 fragments onto a generation 2 core. This method is more rapid than the traditional monomer by monomer synthesis of dendrimers. Moreover, the approach allows for maintaining a variable stereochemistry. While this would be a rapid way to build dendrimers, the authors reported that in this specific case the subsequent formation of what would be a generation 8 dendrimer failed, citing the intense steric shielding of the amino branching points as possible cause. As proof of concept that these dendrimers built by an accelerated synthesis lend themselves to biomedical applications, the authors covalently coupled the generation 4 dendrimer successfully to porphyrin. 

Dendritic p(*L*-Glu) structures have been considered for drug delivery purposes. Starting from a polyhedral oligomeric silsesquioxane (POSS) core Gu et al. constructed a p(*L*-Glu) dendrimer [75,76]. The POSS core was synthesized from (3-aminopropyl)triethoxysilane and its terminal amino groups were converted with a diacid to produce a peripheral shell of carboxyl groups. These carboxyl groups initiated the formation of the 1st generation of the dendritic p(*L*-Glu). Several dendritic generations were built and the last generation was decorated with a targeting moiety (biotin) and a drug (Doxorubicin). Doxorubicin was covalently attached using pH sensitive hydrazine linkers, thereby providing the delivery system with a pH-triggered release capacity. The dendritic structures with biotin showed a much higher cellular uptake and a pH-activated drug release compared to those without biotin. 

Dendritic structures based on p(*L*-Glu) might have a positive effect on the effectiveness of adamantanes, a class of anti-influenza drugs. Adamantanes, unfortunately, were compromised as influenza viruses developed resistance to them. It has been shown however, that the potency of the drug could be recovered by coupling it to a linear p(*L*-Glu) chain [77]. On the other hand, Ranganathan and Kurur had shown already in 1997 that adamantane can be used as the core for the formation of p(*L*-Glu) and p(*L*-Aspartate), p(*L*-Asp), dendrimers [6]. Starting with a bifunctional carboxyl-terminated adamantane core up to three generations of p(*L*-Glu) and p(*L*-Asp) dendrimers were built. As observed for the synthesis of linear PAAs, p(*L*-Asp) dendrimers were harder to synthesize and were obtained in lower yields than their p(*L*-Glu) counterparts.

Interestingly, p(*L*-Glu)-based dendrimers have been investigated for gene delivery purposes as well, with the intent to improve the low transfection efficiency observed for some polycationic delivery systems, caused by serum inhibition and inherent cytotoxicity. Wu et al. developed a dendrimer, that can be envisioned as a three-layer onion with the middle layer being oligo(*L*-Glu) chain [70]. The inner core is a 2nd generation poly(amidoamine) dendrimer; its amino groups were used as initiators for the ROP of benzyl-*L*-glutamate N-carboxyanhydride, introducing an oligo(*L*-Glu) sphere (DP~5) around the poly(amidoamine) core. As cationic charges are, however, necessary for DNA complexation, terminal carboxyl groups on the outer surface were decorated with another generation-1 poly(amidoamine), thus introducing cationic charges on the outside that are highly concentrated and localized. The construct showed good DNA condensation and negligible cytotoxicity. The oligo(*L*-Glu)-augmented dendrimers showed successful DNA binding and condensation with polyplex sizes ranging between 90 and 170 nm depending on the N/P ratio and satisfactory transfection efficiency. Most importantly, no inhibitory serum effects were observed. 

It should be noted that even though dendrimers have been studied for more than 40 years for the delivery of genes, drugs and other biomedically relevant moieties, they have not translated in large scale into the clinic [78]. There are two ways of incorporating drugs and other medically relevant compounds into dendrimers: (i) physical entrapment and (ii) conjugation to functional groups. In case of physical entrapment other, more readily synthesizable and less costly carriers are available, such as nanoparticles, liposomes and polymeric micelles. The special features that dendrimers offer are not utilized when an active compound gets physically entrapped. Dendrimers provide a large and controllable number of functional surface groups, however, only a small amount of drug molecules can be attached to the dendrimer, otherwise the physical properties, e.g., solubility of the dendrimer, will be changed. Hence, the question of how many functional groups are needed is more relevant than of how many of them can be produced. The addition of active compounds to dendrimers leads to multidisperse adducts that are difficult to reproduce, which lowers their chances to pass regulatory standards. The attractiveness of dendrimers to the biomedical/pharmaceutical industry could be enhanced by further pursuing their antiviral capacities or by producing multifunctional dendritic structures and utilizing internal functional groups for the conjugation of drugs. That such constructs can be synthesized has been shown by Vinogradov [71].

## 4. Poly(amino acid)-Based Dendrigrafts

Dendrigrafts, sometimes called dendritic graft polypeptides [79], are the youngest member of the highly branched PAA family. Research into PAA dendrigrafts only emerged in the early 2000s. A significant amount of initial work on dendrigrafts stems from the CNRS polymer research center in Montpellier, France [80,81,82,83,84]. The main advantage of dendrigrafts over dendrimers is their straightforward synthesis by ROP of respective amino acid *N*-carboxyanhydrides. Dendrigrafts are generated from linear polymers, which are considered to be Generation 1. After the protective groups are removed from terminal functional groups, they function as macroinitiators to initiate the formation of the second generation, again by ROP of the respective amino acid *N*-carboxyanhydride. Unlike in dendrimer synthesis, where every reaction step expands the molecule by one AB_2_ monomer, ROPs yield short PAA chains. After another deprotection step a generation 3 can be synthesized, Figure 1. The synthesis of dendrigrafts is not as controlled as that of dendrimers. For example, functional groups that were available but did not participate in the formation of Generation 2 may initiate a polymerization when Generation 3 is synthesized. Since oligo-size chains are attached per reactions step, not monomeric building blocks, the polymer grows rapidly and very high molecular weights are obtained in only two or three polymerization steps, i.e., generations. Thus, structures are formed that have a lesser structural rigidity and order. On the other hand, producing multi-gram quantities is readily possible with this technique. Chain length must be chosen conscientiously as each repeat unit theoretically acts as an initiator and molecular weight increases rapidly from generation to generation. Detailed polydispersity studies by Boiteau et al. using capillary electrophoresis and size exclusion chromatography showed polydispersity indexes ranging between 1.20 and 1.50 [80,85]. So far, most of the work was done on p(*L*-Lys)-based dendrigrafts, hence ε-amino groups form the branching points from which individual generations propagate. 

Dendrigraft synthesis leads to highly branched polymers with a rather large number of peripheral, charged and reactive groups already present in the second or third generation. The number of reactive groups is determined by the length of the linear chains that constitute the terminal generation. If these chains are comparatively long, which can be controlled by the monomer-to-initiator ratio, then a high charge density is achieved. The chain length within individual generations also determines the “softness” of the construct. The longer the chains are the more flexibility, mobility and adaptability to (biological) ligands can be achieved. This structural flexibility seems to be advantageous in the interaction of dendrigrafts with biological molecules even though this point has not yet been clinically proven. Dendrigrafts and hyperbranched polymers are structurally complex and a plethora of structural variants can be synthesized by simply varying synthetic conditions. Thus, a large number of polymers can be synthesized varying in composition, charge density, degree of polymerization and branching ratio. Studying the impact of all these molecular variations would be a complex task, but molecular dynamics simulations have been applied to second to fourth generation p(*L*-Lys) dendrigrafts [86]. The results showed that an enormous number of conformational states exist indicating a pronounced conformational plasticity [87]. This ability to adaptably display positive charges to anionic ligands may be the reason for the strong and efficient binding observed between dendrigrafts and biologically relevant molecules. 

The differences between dendrimers and dendrigrafts is obvious when their hydrodynamic behaviors are considered. Compared to dendrimers dendrigrafts are much softer and more flexible and they exhibit significant changes in the solvent-dependent hydrodynamic dimensions. It was shown that the hydrodynamic volumes of p(*L*-Lys) dendrigrafts were contracted by a factor of eight in DMF compared to an aqueous solution [88].

Just like for dendrimers, the main body of research work focuses on dendrigrafts derived from p(*L*-Lys) [89]. They share their main chemical and biological characteristics with their dendrimer counterparts. Those characteristics include an overall good biocompatibility, a good solubility in water [80], susceptibility to the action of proteases and peptidases [90], a comparatively low cytotoxicity, they are rather non-immunogenic [91] and carry an inherent biocidal activity, because of their polycationic structure. 

Initially p(*L*-Lys)-based dendrigrafts were investigated as carriers for antibody production because it was observed that a 3rd generation p(*L*-Lys) dendrigraft by itself was non-immunogenic, which advantageously distinguished it from other carriers for peptide antigens [91]. Given the unique interaction between the rather soft and flexible dendrigrafts and proteins and exploiting the surface functional groups for further modification, p(*L*-Lys) dendrigrafts were evaluated as release systems for insulin [92]. Here, the dendrigraft surface was decorated with arginine residues, as arginine-rich structures improve the cell membrane permeability, in particular for (drug) molecules that are taken up poorly. Sideratou et al. showed that insulin association efficiency exceeded 99% in a 2nd generation p(*L*-Lys) dendrigraft [92]. The insulin release was related to the concentration of arginine end groups on the construct; higher arginine concentrations led to a slower insulin release over a longer period of time when released into simulated intestinal fluid.

Linear p(*L*-Lys) has been used widely as coating to induce nerve cell adhesion, e.g., for fibrous scaffolds. However, linear p(*L*-Lys) conveys a low charge density when it wraps around fibrous or filamentous materials. Replacing linear p(*L*-Lys) with p(*L*-Lys) dendrigrafts led to constructs where the coating of poly(glycolic acid) fibers, slated as nerve cell scaffolds, was less stable and the degree of association was lower. This can probably be attributed to the three-dimensional nature of dendrigrafts and their considerable flexibility, which does not allow them to actually wrap themselves around fibers. However, when keeping the molecular weight of linear p(*L*-Lys) and a 4th generation p(*L*-Lys) dendrigraft similar, more neuronal cells adhered to the scaffold coated with dendrigrafts [93]. This can be explained by the fact that dendrigrafts positioned along the fibrous scaffold provide focal points with a high cationic charge density, which promote neuronal growth.

A variation on dendrigrafts, termed “denpol”, see Figure 7, has been proposed by Zeng et al. where dendritic structures have been introduced along a peptidic backbone obtained by step-growth polymerization of dicysteine and *L*-Lysine [94]. A p(*L*-Lys) dendron was then grown onto this backbone to a desired generation and the outer surface of these dendrons was decorated with amino acids that introduced a hydrophobic or hydrophilic character. The construct proved useful for the delivery of siRNA as the flexible backbone to which the dendrons were attached provided the “softness” and flexibility needed for effective binding of siRNA and the spatial separation of the dendrons along the chain aided in forming stable complexes. Since the outer surface of the dendrons can be decorated with different (amino acid) moieties various aspects of the cellular uptake can be addressed simultaneously. Moreover, the disulfide linkages in the backbone respond to the reducing environment of the cytoplasm. 

Since the physical properties of dendrigrafts are not that different from dendrimers, that is, three-dimensional structures with either a high surface charge, or a chemically decorated surface, the biomedical applications for dendrigrafts do not vary that much from those for dendrimers. The constructs were first considered for gene delivery [95,96,97,98,99,100,101] and later the focus shifted towards drug delivery [102,103,104,105,106,107,108,109,110]. Limited work on the use of dendrigrafts in biomedical imaging [111] was also reported. Niche studies have revealed other interesting properties inherent to dendrigrafts, e.g., Vial et al. found that a second-generation dendrigraft of p(*L*-Lys) neutralizes the anticoagulant activity of unfractionated heparin and low-molecular-weight heparin and could replace protamine, a drug that is typically used after cardiac bypass surgery or to treat perfuse bleeding and accidental heparin overdoses but poses several other medical risks [112]. Moreover, using the same arguments made above for dendrimers, p(*L*-Lys) and p(*L*-Glu) dendrigrafts have been investigated for inhalable gene delivery systems [113].

## 5. Hyperbranched Poly(amino acid)s

Paul Flory [114] hypothesized on the complexities of highly branched, three dimensional polymers derived from trifunctional monomers as early as 1941, but it took almost half a century until experimental work started and Kricheldorf showed that a branched polycondensate from hydroxybenzoic acid and 3-(trimethylsiloxy)benzoyl chloride could be synthesized without any crosslinking [115]. Kim and Webster [116] were the first to use the term hyperbranched polymer in 1990 when they synthesized a hyperbranched poly(phenylene) and Menz and Chapman [117] were among the first to investigate the synthesis of hyperbranched p(*L*-Lys). Research into p(*L*-Lys)-based dendrimers had shown that higher generation dendrimers are more efficient in gene delivery than linear polymers [118], however, the chemical synthesis of higher generation dendrimers is very challenging. Hyperbranched polymers, including hyperbranched PAAs, are considered alternatives to dendrimers and while the discussion of whether they are superior, equal or inferior in their biomedical applications to dendrimers has not yet been decided, they clearly hold the advantage when it comes to industrial manufacturing, as their synthesis is readily scalable. They can be produced quickly and in appreciable amounts [119]. All research on hyperbranched polymers to date focuses on p(*L*-Lys) hyperbranched polymers. 

Hyperbranched p(*L*-Lys) can be readily synthesized in the melt or aqueous solution of *L*-Lys hydrochloride in the presence of a strong base, e.g., KOH, NaOH, LiOH or CsOH in a one-pot synthesis using amidation catalysts based either on zirconium, titanium, antimony and 3-pyridine boronic acid, Figure 8. The entire protection/deprotection chemistry that is key to dendrimer syntheses is here omitted. Since the reactivities of the two amino functions in *L*-Lys are different, with the N^ε^ amino group being more reactive, thermal polymerization yields hyperbranched polymers with a larger amount of N^ε^ linked structural elements. Polymers of acceptable molecular weights form within 24 h, but as condensation reactions continue to proceed, the molecular weight and polydispersity continue to increase, with polydispersities ranging between 2.5 and 4.0 [120,121]. 

The more readily synthesizable hyperbranched polymers were investigated for the same purpose as their dendrimer relatives, i.e., non-viral gene delivery vehicles. Their three-dimensional structure that derives from a high, but less organized, degree of branching provides them with a unique structure that is not found in dendrimers and dendrigrafts. Hyperbranched polymers are not monodisperse, and they do display what has been referred to as ‘structural defects’. However, these structural imperfections provide hyperbranched polymers with a unique 3D structure that makes them less organized and “softer” with a large number of intramolecular cavities that are advantageous for accommodating cargo. The large number of functional groups distributed throughout the polymer, and not localized at the corona of the three-dimensional object, can be utilized in subsequent modifications bringing derivatizations deeper into the structure than they would be the case for dendrimers. 

Thermal polymerizations used in the synthesis of hyperbranched polymers are not controlled for molecular architecture and molecular weight. However, these polymerizations can be modulated by the addition of small molecules that contain carboxyl functions or carboxyl and amino functions or are based on copolymerizations with N^ε^-protected *L*-Lysine [120]. These modulators can affect the polymerization kinetics, the molecular weight and the structure of the hyperbranched polymer. For example, the ratio of α- to ε- connections in hyperbranched p(*L*-Lys) can be influenced by the addition of o-vanillin or a copolymerization with ε-benzylidene-*L*-lysine. These modulators promote the polymerization via the less reactive N^α^-amino group. The addition of o-vanillin led to reduced molecular weights and significant structural changes were observed in favor of fractions connected via the N^α^-amino group. Similarly, the use of the N^ε^-protected amino group was expected to shift the polymerization towards the less reactive N^α^-amino group. Using ε-benzylidene-*L*-Lysine as a comonomer in the p(*L*-Lys) synthesis caused such a shift in structure towards more N^α^-connections when the mole fraction of the ε-benzylidene-*L*-lysine was above 25%; the molecular weight and polymerization kinetics were affected as well. The addition of α-amino-ε-caprolactam had, however, no impact. 

Another way to overcome the limited control over molecular weight and polydispersities was investigated by Frey et al. who conducted computer simulations of the kinetic processes and suggested the ‘slow addition technique’, that is, monomer is added to the reaction throughout the entire polymerization process [122]. This approach was experimentally tested by Menz and Chapman [117] and it was found that higher molecular weights were achieved. Another attempt at controlling the syntheses of hyperbranched PAAs was made by Li and Dong who built upon the ROP of *L*-Lys-NCA [123]. The ROP of a photocaged N^6^-(o-nitrobenzyloxycarbonyl)-*L*-Lys NCA was photo-triggered by irradiation with UV light at λ = 365 nm. The o-nitrobenzyloxycarbonyl group rearranges upon light irradiation and converts the N^6^-(o-nitrobenzyloxycarbonyl)-*L*-Lys NCA into an “inimer” (for: initiator + monomer) NCA that has a free amino group as nitrobenzaldehyde is removed. The polymerization proceeds at room temperature without the need for catalysts. By fine-tuning the UV irradiation time and intensity hyperbranched poly(*L*-Lys) was produced with controllable molecular weights and controllable degrees of branching that ranged between 0.09 and 0.60 [123]. 

The majority of the research into p(*L*-Lys) hyperbranched polymers came from Klok’s group [124,125,126,127,128,129,130]. Because of its cationic nature hyperbranched p(*L*-Lys) is expected to act as a gene delivery vehicle. Testing hyperbranched poly(*L*-Lys) as transfection agent for the production of IgG antibodies, in transient gene expression as well as in the transfection of primary mammalian cells, revealed that hyperbranched poly(*L*-Lys) could perform on a level comparable to poly(ethylene imine), PEI, in the production of recombinant proteins [128]. As observed in other polycationic delivery systems, the yield of recombinant protein increased with the molecular weight of the carrier. A significant difference between PEI and hyperbranched p(*L*-Lys) is, however, that the latter is susceptible to proteolytic degradation. This biodegradability is a significant advantage as the cumulative cytotoxicity of the transfection agent is kept at a minimum. On the other hand, premature degradation of the p(*L*-Lys) hyperbranched polymer can render the gene delivery process inefficient. However, when considering the recovery of recombinant proteins in scaled-up production settings the proteolytic degradability is a distinct advantage. 

When compared to linear and dendritic p(*L*-Lys), hyperbranched polymers performed superiorly as nucleic acid delivery vehicles [127]. At comparable molecular weights hyperbranched p(*L*-Lys) showed a higher transfection efficiency. When studying the amount of enhanced green fluorescent protein, eGFP expressed, it was found that the percentage of eGFP positive cells after 24 h of transfection was about 10% for linear and dendritic p(*L*-Lys) at their highest molecular weights of about M_w_ = 18,000 Da and 30% for a hyperbranched polymer of comparable molecular weight. Increasing the molecular weight of the hyperbranched p(*L*-Lys) to 230,000 Da, which can be readily achieved, yielded up to 60% of eGFP positive cells in primary mammalian cells. Even more distinct is the difference in transfection ability when considering the production of IgG in primary mammalian cells; using linear and dendritic p(*L*-Lys) delivery vehicles, IgG yields of about 0.075 mg/L (linear) and 0.05 mg/L (dendritic) were achieved. Using a hyperbranched p(*L*-Lys) with comparable molecular weight yielded 1.0 mg/L and the yield increased to 2.0 mg/L for higher molecular weight hyperbranched p(*L*-Lys). These major differences can be explained by studying the structure-property relationships of these polymers and the cell uptake and trafficking of polyplexes formed with DNA. Polyplexes based on hyperbranched p(*L*-Lys) had a higher binding affinity to the external cell membrane, which is probably due the softer, more malleable structure. Moreover, upon exceeding an N:P ratio of 3:1, it was observed that a larger fraction of free polymer coexists with the polyplex. Again, this is most likely a result of the less organized structure of the hyperbranched polymers. A much higher pH buffering capacity was observed for the hyperbranched polymer, which affects the pH in the late endosomal and lysosomal compartments. These combined factors are thought to be key to the higher transfection efficiency of hyperbranched p(*L*-Lys) gene delivery vehicles. However, it needs to be kept in mind that linear p(*L*-Lys) displays a significantly lower acute cytotoxicity than hyperbranched and dendritic polymers of comparable molecular weights. Moreover, branched polymers induce a significantly higher degree of apoptosis than linear polymers [128]. These differences are probably the result of the higher proteolytic stability of the branched polymers. These results indicate, as for all polycationic delivery vehicles, that a balance must be kept between the biocompatibility of the vector and its efficiency.

Studies by Alazzo et al. [131,132] showed that the efficiency of gene delivery systems depends on the use of comonomers among other factors. Their molar content, placement and sequence within the structure can distinctly alter the efficiency of the delivery vehicle. In an effort to reduce the cytotoxicity of branched p(*L*-Lys) delivery structures, Alazzo et al. performed the thermal polycondensation of *L*-Lys HCl in the presence of 10 to 40 mole% of histidine [131]. In addition to a reduced cytotoxicity, histidine derivatives of p(*L*-Lys) were also expected to enhance the transfection efficiency as histidine can become protonated thereby acting as a “proton sponge”, which would be advantageous in the endosomal compartments. The molecular weights of the histidinylated hyperbranched p(*L*-Lys) ranged between 22 and 87 kDa with PDIs between 1.5 and 1.9, Figure 9. Contrary to the expectations, and despite the fact that histidine would provide an enhanced buffer capacity the hyperbranched copolymers did not yield a higher transfection efficiency. A detailed analysis of the structures revealed that histidine residues modulated the structure by competing with the more reactive N^ε^-amino groups of *L*-Lys, thereby directing the polymerization towards the N^α^-amino groups, as shown by Scholl et al. [120], see above. The resulting polymeric structure was more rigid as shown by the degree of branching and the glass transition temperatures, which increased from 57 to 65 °C for hyperbranched p(*L*-Lys) to 107 to 113 °C for histidinylated hyperbranched p(*L*-Lys) with 40 mole% L-histidine. It was observed that the histidinylated copolymers displayed an overall lower surface charge and had a higher tendency to aggregate. The effectiveness of these hyperbranched polycation gene systems was investigated by metabolomic techniques studying the downregulation of metabolites associated with glycolysis and the tricarboxylic acid cycle and the induction of oxidative stress in lung cancer cell lines. The results indicated that, polyplexes based on the p(*L*-Lys) and the histidinylated p(*L*-Lys) hyperbranched polymers were better tolerated than PEI polyplexes [132]. 

While the bulk of research into p(*L*-Lys) hyperbranched polymers focuses on gene delivery applications, Zu et al. evaluated them as delivery vehicles for Gd(III) MRI contrast agents [133]. The multitude of functional groups allows for their conjugation, here with folic acid, to target the folate receptors on tumors. The contrast agent was biocompatible and compared to commercially available reagents it exhibited three times higher longitudinal relaxivity values. The cellular uptake was significantly enhanced for the folic acid carrying contrast agent and in vivo experiments (mice) showed a significant signal intensity enhancement in the tumor region. 

## 6. Summary

Dendrimers, dendrigrafts and hyperbranched polymers based on PAAs were examined for their potential use as non-viral vectors in gene and drug delivery applications. The use of dendrimers is encouraged and justified if the highly defined structure with a pre-determined number of functional groups can be exploited, i.e., the covalent attachment of biomedical compounds to the dendrimer corona. Their use in antiviral application has been advanced to marketability with the dendrimer platform SPL7013 produced by to Starpharma, which is currently of pertinent relevance with respect to the coronavirus and the pandemic that ensued. Future similar viral outbreaks must be anticipated, and preventive measures, such as dendrimer-based antivirals might prevent another escalation to the stage of another pandemic. The entrance of dendrimers into the gene delivery realm seems however to be hampered, mostly due to the laborious dendrimer synthesis and the fact that comparable transfection results can be achieved with delivery systems that are more readily synthesizable. It has been shown that flexibility in the delivery system is more important than a high charge density. This opened the way for the study of the structurally less organized dendrigrafts and hyperbranched polymers for biomedical applications. 

Hyperbranched polymers show the most flexibility due to their structural diversity. While this flexibility is an advantage in terms of accommodating cargo, it is also a regulatory challenge, as the structural multiplicity, which is potentiated when complexed with drugs or nucleic acids, is difficult to define and regulate. Dendrigrafts and hyperbranched polymers have not yet been introduced into the clinic. The renewed interest in antivirals and the discoveries anticipated in this sector over the next few years may also be translated into the future gene and drug delivery market.

## Figures and Tables

**Figure 1 nanomaterials-11-01119-f001:**
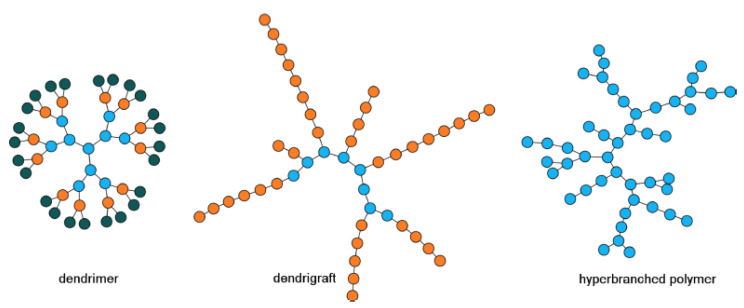
Highly branched polymers: (**left**): 3rd generation dendrimer, (**middle**): 2nd generation dendrigraft, (**right**): hyperbranched polymer; reproduced with permission from Francoia and Vial [21]

**Figure 2 nanomaterials-11-01119-f002:**
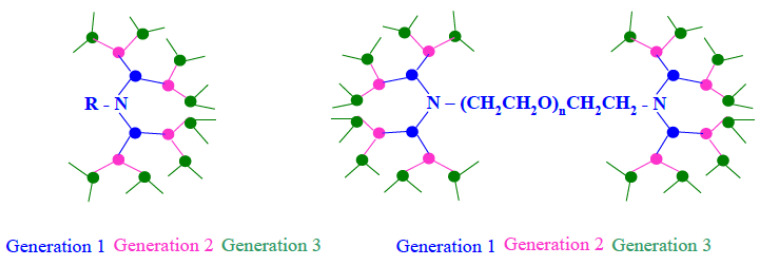
Formation of a 3rd generation dendrimer from a mono-amino initiator, each dot represents an asymmetrical AB_2_ amino acid molecule. (**Left**): R is the initiator and can be a small molecule resulting in a compact spherical dendrimer or a water-soluble biocompatibilizing polymer, e.g., PEG, yielding a PEG-anchored dendrimer. (**Right**): a dumbbell shaped dendrimer forms from a bifunctional polymeric initiator.

**Figure 3 nanomaterials-11-01119-f003:**
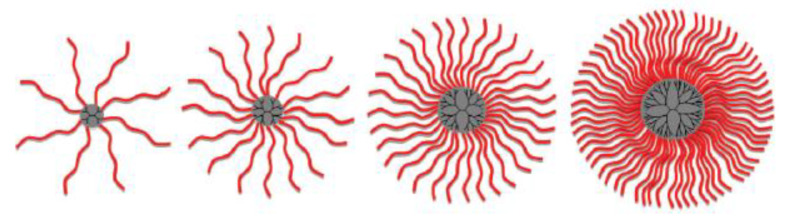
Star-shaped dendrimers synthesized by ring-opening polymerization of ε-carbobenzyloxy-*L*-lysine N-carboxyanhydride using generation 2nd to 5th generation dendritic macroinitiators. Depending on the generation of macroinitiator, the dendritic structure has a pre-determined number of polymeric “arms” of a rather constant chain length; reproduced with permission from Byrne et al. [29].

**Figure 4 nanomaterials-11-01119-f004:**
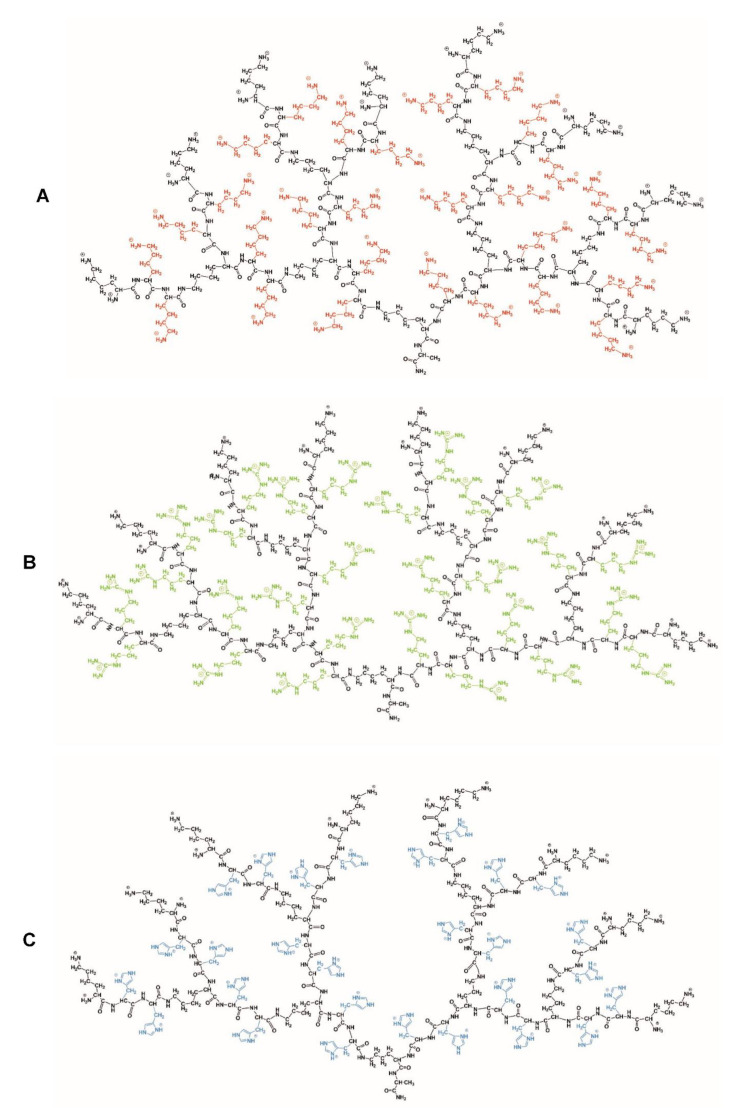
Dendrimer consisting of p(*L*-Lys) (**A**), and augmented with *L-*arginine (**B**) and *L-*histidine (C). The inclusion of amino acids other than L-lysine influences the zeta potential of the construct, and the cytotoxicity decreases with decreasing zeta potential; reproduced from Gorzkiewicz et al. [36].

**Figure 5 nanomaterials-11-01119-f005:**
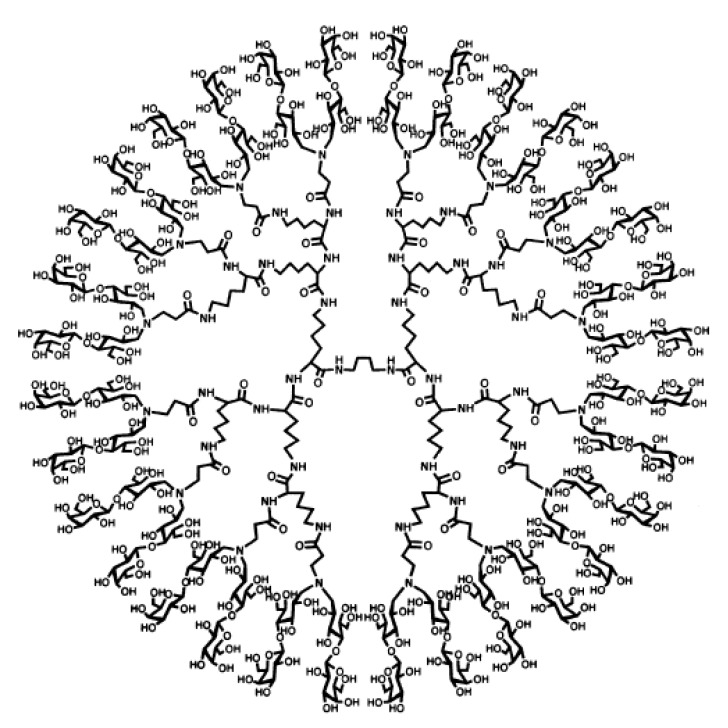
Oligosaccharide-decorated dendrimer, here maltose is attached via an alanine linker to a 3rd generation p(*L*-Lys) dendrimer; reproduced with permission from Baigude et al. [59].

**Figure 6 nanomaterials-11-01119-f006:**
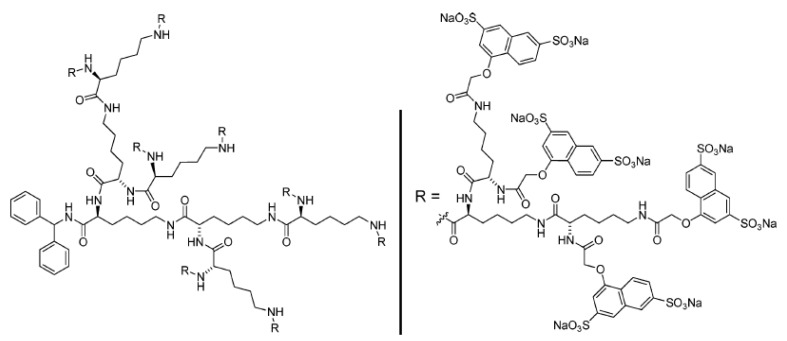
Starpharma’s SPL7013, also known as astodrimer sodium, a 4th generation p(*L*-Lys) dendrimer with an anionic surface charge. This product shows promise in the treatment of coronavirus disease, reproduced with permission from McCarthy et al. [68].

**Figure 7 nanomaterials-11-01119-f007:**
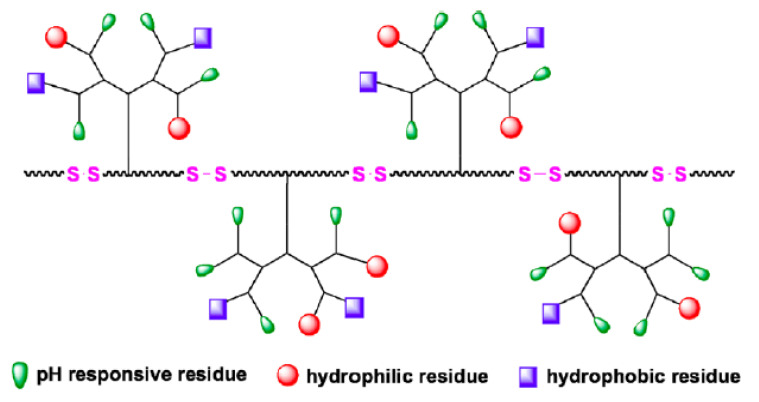
“Denpol”: A dendritic structure has been grown from a peptidic backbone that was produced by Scheme [94].

**Figure 8 nanomaterials-11-01119-f008:**
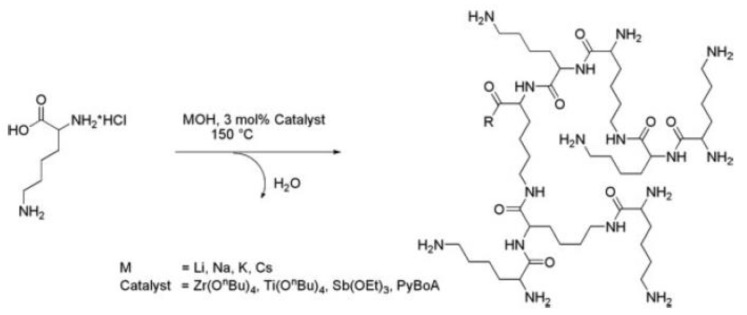
Thermal synthesis of hyperbranched p(*L*-Lys) in the melt in a one-pot synthesis from *L*-lysine hydrochloride, reproduced with permission from Scholl et al. [121].

**Figure 9 nanomaterials-11-01119-f009:**
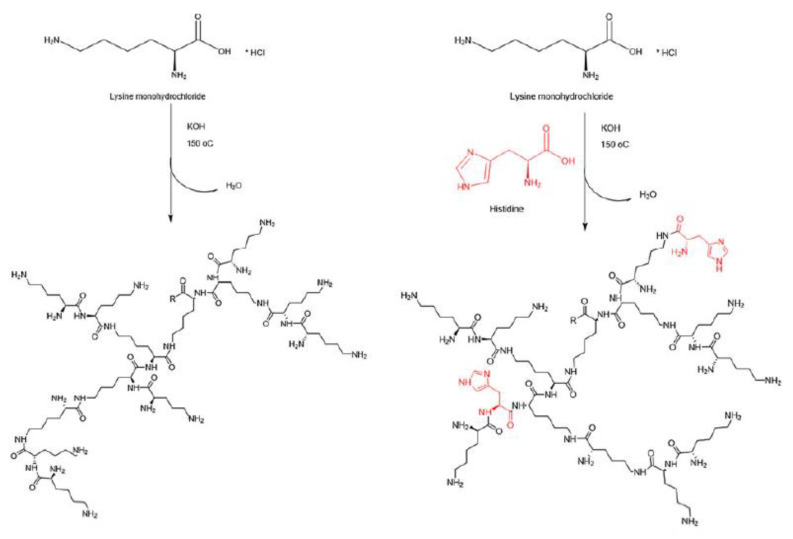
Thermal polycondensation of *L*-lysine HCl in the absence (**left**) and presence of histidine (**right**). Histidine is copolymerized into the hyperbranched structure and reduces the flexibility of the branches; reproduced from Alazzo et al. [131].

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
