# Peer review of "Highly Branched Polymers Based on Poly(amino acid)s for Biomedical Application"

_nanomaterials, 2021, doi:10.3390/nano11051119_

Round 1

Reviewer 1 Report

The review “Highly Branched Polymers Based on Poly (amino acids) s for Biomedical Application” presented by Marisa Thompson and Carmen Scholz is written on a very relevant topic. The material is presented in a logical manner and is easy to read. However, the submitted manuscript has a number of questions and suggestions:

  1. In the introduction, it is necessary to focus on (1, 2 sentences) the nano-sized packaging of the obtained polymers. The subject of the journal is still nanomaterials, so it is necessary to focus on this.
  2. Please make your links according to the Nanomaterials rules. Unselected links in the text confuse the reader.
  3. The quality of the pictures should be improved.
  4. Line 68, 69: The authors describe the synthesis of hyperbranched polymers by one step thermal polymerization. At the same time, there is no control over the resulting polymer structure.

Further (lines 73, 74), when describing dendrigrafts, the authors do not give methods of their preparation and claim that they are not monodisperse.

What kind of dispersion are we talking about? By what methods was it shown (DLS?). What are the synthesis methods dendrigrafts? What is the molecular size distribution of hyperbranched polymers? Several sentences need to be added.

  1. In the introduction, it is necessary to give examples of hyperbranched frameworks that are similar in properties to amide polymers (for example – polyesters or PLA- thermoplastic polyester).
  2. Line 155: when specifying the sizes of the resulting particles, it is necessary to indicate the method by which these sizes were obtained (it would be great if the authors gave TEM or SEM images, if any).
  3. Line 173, 174: "Compared to linear p (L-Lys) the star shaped polymers performed superiorly, producing tightly packed polyplexes at significantly lower N: P ratios."

Would like to know the dimensions of the resulting polyplexes. What methods of the ball is shown the star-shaped form of dendrimers. If there are TEM images, please provide!

  1. Line 196-198: Discuss the difference in zeta-potentials in more detail. Give the values of the zeta-potentials. How can toxicity be related to the value of zeta-potentials? Discuss this mechanism.
  2. Lines 443, 443: Give the method by which the polydispersity index was determined.

The reviewer recommends accepting the manuscript after revision.

Author Response

  1. A reference to the importance of nano-sized structures has been added in the introduction.
  2. We are sorry, but we are not sure what is meant by ‘unselected links”
  3. The quality of the figures has been improved.
  4. Yes, this is correct, thermal one-pot polymerizations of hyperbranched polymers do not allow for any control over the polymer architecture. Equally, the synthesis of dendrigrafts, described in chapter: Poly(amino acid)-based Dendrigrafts, page 14,15, does not yield monodisperse constructs either, again by virtue of their syntheses.  In fact they cannot be monodisperse because the functional terminal groups in the side of a linear poly(amino acid) act as initiator for the synthesis of the second generation of linear poly(amino acid).  Then again, their terminal functional side groups act as initiator in the subsequent step.  As all generations are linear polymers, monodispersity cannot be achieved.   The degrees of polydispersities achieved by various researchers in their specific investigations varies.  As an example, the work of Pascal and Collet was quoted here as their group investigated the polydispersities of dendrigrafts in great detail and a clarifying remark has been added.  But the original literature should always be consulted for more specific information on each dendrigraft. 
  1. We believe that including hyperbranched polyesters will be outside the scope of this manuscript. Lactic acid is not an AB2 monomer, typically hyperbranched polyesters are derivatives of glycerols or cellulose, i.e. the chemistries are significantly different from the PAAs discussed here.

6 and 7.  This is a review article, which should give the reader an introduction and insight into the topic; detailed information is available from the original literature.   Providing such detailed information for one polymer system (e.g. TEM, SEM) would unfairly emphasize this particular work.  Again, the reader is referred to the original work. 

We do not quite understand what is meant by “What methods of the ball is shown the star-shaped form of dendrimers.”

8. It is well known that the zeta potential of a polymer determines its cytotoxicity as the cationic charges interfere with the cell wall of microorganisms. This process is used extensively in antimicrobial polymers and antimicrobial polymer coatings.  For more information on this subject the reviewer is referred to the large body of work provided, for example, by Kenichi Kuroda.  Since this is a review article, its purpose is to showcase the various and possible applications of PAAs.  Going into mechanistic (or analytical) details outside the structure-application focus of this manuscript, will go beyond the scope of this review article.

9. Unfortunately, we do not have access to a line-numbered ms, but again such detail information as methods used to determine the polydispersity index are readily available in the original literature.

Reviewer 2 Report

This review is in my opinion interesting and could be accepted for publication in Nanomaterials. The main disadvantage of this ms. is unacceptable, very poor quality of drawings (figures, schemes). All of them should be re-drawn. Citations in the body text should be given as superscripts.

Minor comments:

  1. Lack of Greek letters in the body text, like α, β, ε, etc.  A strange symbol of unknown meaning is present instead.
  2. Cit. 117. Replace "Chapmun" by "Chapman"
  3. Cit. 68. This citation does not concern studies on any anti-SARS CoV-2 therapy.

Author Response

  1. The quality of the figures has been improved.
  2. We are sorry about this problem, but we cannot find strange symbols; all Greek letters are in order in our copy. We hope this issue will be addressed in the type-setting.
  3. Thank you for catching this typo, it has been corrected.
  4. The inhibitory action of SPL7013 has been clarified.

Reviewer 3 Report

In this paper, Scholz and Thompson review the use of highly branched polymers based on poly(amino acid)s for biomedical application. Specifically, the review covers the synthesis of organized dendrimers, dendrigrafts and hyperbranched polymers and their potential application as drugs and genes delivery systems and antiviral compounds. The review is well-written and structured, and references are well-balanced. Therefore, I think the manuscript should be accepted for publication in Nanomaterials after the following minor comments have been addressed.

  1. Please change all reference numbers to [number] throughout the manuscript following the format for MDPI articles.
  2. Authors should change most of figures since the quality is not good enough for publication.
  3. Please check references format since some of them do not follow MDPI format.

Author Response

  1. Reference numbers have been changed to match MDPI style.
  2. The quality of the figures has been improved.
  3. The missing information has been added.

Round 2

Reviewer 3 Report

Authors have addressed all comments and the manuscript is suitable for publication in Nanomaterials